# High-Light-Induced Degradation of Photosystem II Subunits’ Involvement in the Albino Phenotype in Tea Plants

**DOI:** 10.3390/ijms23158522

**Published:** 2022-07-31

**Authors:** Wen-He Cai, Xin-Qiang Zheng, Yue-Rong Liang

**Affiliations:** Tea Research Institute, Zhejiang University, #866, Yuhangtang Road, Hangzhou 310058, China; caiwenhe@zju.edu.cn (W.-H.C.); xqzheng@zju.edu.cn (X.-Q.Z.)

**Keywords:** *Camellia sinensis*, albino tea, photosystem II, photodamage, thylakoid

## Abstract

The light-sensitive (LS) albino tea plant grows albinic shoots lacking chlorophylls (Chls) under high-light (HL) conditions, and the albinic shoots re-green under low light (LL) conditions. The albinic shoots contain a high level of amino acids and are preferential materials for processing quality green tea. The young plants of the albino tea cultivars are difficult to be cultivated owing to lacking Chls. The mechanisms of the tea leaf bleaching and re-greening are unknown. We detected the activity and composition of photosystem II (PSII) subunits in LS albino tea cultivar “Huangjinya” (HJY), with a normal green-leaf cultivar “Jinxuan” (JX) as control so as to find the relationship of PSII impairment to the albino phenotype in tea. The PSII of HJY is more vulnerable to HL-stress than JX. HL-induced degradation of PSII subunits CP43, CP47, PsbP, PsbR. and light-harvest chlorophyll–protein complexes led to the exposure and degradation of D1 and D2, in which partial fragments of the degraded subunits were crosslinked to form larger aggregates. Two copies of subunits PsbO, psbN, and Lhcb1 were expressed in response to HL stress. The cDNA sequencing of CP43 shows that there is no difference in sequences of PsbC cDNA and putative amino acids of CP43 between HJY and JX. The de novo synthesis and/or repair of PSII subunits is considered to be involved in the impairment of PSII complexes, and the latter played a predominant role in the albino phenotype in the LS albino tea plant.

## 1. Introduction

Albino tea plants (*Camellia sinensis* (L.) O. Kuntze) are mutants that grow albinic shoots under particular environmental conditions [1,2]. There are two types of albino tea plants grown in production, i.e., a temperature-sensitive (TS) one and a light-sensitive (LS) one [3,4]. The former grows albinic leaves in white color when the environmental temperature is below 20 °C during early spring season but grows green leaves during summer and autumn seasons when temperature is higher than 20 °C; on the other hand, the latter grows albinic leaves in yellow color under high-light (HL) conditions and grows green leaves under lower-light (LL) conditions [3]. Both show the albino phenotypes owing to their deficiency of chlorophylls (Chls), and their albinic tea leaves contain a high level of amino acids, which are excellent materials for processing quality green tea [5,6]. Many attempts were made to reveal the albino mechanism of TS albino tea plant [7,8,9]. However, studies on LS albino tea are rare [9], and the albino mechanism of the LS albino tea plants remains to be investigated [5].

Amino acids are major components of tea, and their concentration in tea leaves are positively correlated to the sensory quality of green tea [6,7,8]. Tea shoots grown on the LS albino tea plants contain a high level of amino acids, so they are preferential materials for processing quality green tea [9]. However, the young plants of the LS albino tea cultivars are vulnerable to HL stress in summer, and the chemical composition of their tea shoots varies with the environmental light intensity, resulting in tea quality instability [3]. Probing the mechanism underlying the LS albino phenotype of tea plant is of great significance for developing countermeasures to promote the growing the LS tea cultivars and to control the quality of these tea products.

Observation on leaf ultrastructure showed that HL induced impairment of chloroplasts by suppressing the development of grana stacking and thylakoids of LS albino tea plants [10]. Chloroplasts of the leaves grown under HL conditions had fewer starch granules, osmiophilic granules, and thylakoids stacking, whereas those of the leaves grown under shaded LL conditions were fully developed, without abnormity in thylakoid membranes and granular stacking [11]. There was a total of 507 genes showing differential expression levels in response to light changes, most of which were involved in light-harvest protein complex, pigmentation, and protein processing [9,12]. The repressed genes encoding light-harvesting Chl a/b-binding proteins (LHCB) were closely linked to aberrant chloroplast development in the LS albino leaves [12].

Photosystem II (PSII) associates with light-harvesting complexes II (LHCII) to form PSII–LHCII supercomplexes (PSII–LHCII SC). Chls and their binding cofactors are antennas for light-harvesting complex subunits of PSII–LHCII SC and photosystem I (PSI) and their light-harvesting complex I (LHCI) along with the Chl a-binding antenna proteins CP43 and CP47 [13]. In PSII–LHCII SC, PSII reaction center (PSII RC) core dimers D1 and D2 are surrounded by LHCII trimers, which consist of Lhcb1 and Lhcb2 proteins [14]. Light-driven photosynthetic reactions that are carried out in eukaryotic photosynthetic organisms include the capture of light and its conversion to chemical energy. These reactions take place in five major multi-subunit protein complexes in the thylakoid membranes of higher plants: PSII, PSI, Cytb_6_f, ATP synthase, and NDH [14]. Each complex is composed of multiple subunits encoded by both nuclear and chloroplast DNA. PSII is a large pigment–protein complex containing more than 20 subunits that bind more than 100 cofactors, catalyzes light-driven water oxidation, and transfers electrons from water to plastoquinone concomitant with oxygen evolution [15]. PSII is vulnerable because of high-speed turnover of the D1 protein, and it has been shown in numerous studies that PSII is a primary site of photoinhibition [16]. Light is absorbed by pigment cofactors, and excitation energy is transferred among antennae pigments and converted into chemical energy with extremely high efficiency. Plants use light for photosynthesis but can be damaged by the excessive light through the mechanisms of photodamage, photoinhibition, or photobleaching. Photoinhibition refers to light-dependent, irreversible inactivation of PSII RC activity, which can be restored only via the degradation and synthesis of the D1 protein [17]. The activity of PSII or the maximum potential capacity of the photochemical reactions of PSII can be reflected as the ratio of variable fluorescence to maximum fluorescence (Fv/Fm) [18]. Determination of Fv/Fm can yield information on the ability of plants to acclimate and adapt to an environment from the perspectives of PSII activity and photosynthesis, and detecting the corresponding protein expression regulation at different levels can help elucidate the molecular mechanisms, including photodamage and repair processes [18,19].

It is hypothesized that the PSII–LHCII SC proteins in the LS albino tea cultivars are more vulnerable to HL stress than the normal green tea cultivars, resulting in more serious photoinhibition or photodamage of the PSII pigment protein complexes, which lead to deficiency of Chls and the albino phenotype of LS albino tea plants. The damaged PSII pigment protein complexes induced by HL can be replaced with nascent ones to maintain photosynthetic activity when the light intensity is decreased naturally or artificially. Based on this hypothesis, the changes in the chloroplast-encoded proteins, including PSII RC proteins, PSI protein complex, LHC, and PSII activity indicator Fv/Fm, under HL and artificially shaded LL conditions were examined in the present paper, aiming to reveal their relationship to the albino phenotype of LS albino tea plant.

## 2. Results

### 2.1. Effects of HL on PSII Protein Subunits

The leaves grown on plants of LS albino tea cultivar HJY (*Camellia sinensis* cv. Huangjinya) under HL conditions (diurnal sunlight 1200–1900 μmol/m^2^·s) showed a typical foliar albino phenotype and turned green under LL conditions by artificially shading using black net with 30% transmittance. The leaves on normal green-leaf tea cultivar JX (*Camellia sinensis* cv. Jinxuan) showed slightly deeper green color after the shading treatment (Appendix A).

Figure 1 shows the comparison of PSII complex subunits including the Chl a binding antenna proteins CP43 and CP47, PSII RC core proteins D1 and D2, and subunits PsbE, PsbH, PsbI, PsbN, PsbO, PsbP, and PsbR between normal tea cultivar JX and LS albino tea cultivar HJY. In JX, there was no obvious change in the accumulation levels of subunits CP43, CP47, D1, D2, PsbE, PsbH, PsbI, PsbP, and PsbR (Figure 1A–G,J,K) during the 30-day shading treatment with the exception of PsbN, which was weakly detected on the 15th day of shading treatment (Figure 1H). PsbO (~33 kDa) in JX was abundantly accumulated during the 30-day shading treatment, and an additional band (~23 kDa) was weakly observed on the 0th and 6th days (Figure 1I). However, the responses of PSII subunits to the artificially shaded LL treatment in HJY were more complicated than those in JX. First, the accumulations of PsbE, PsbH, and PsbI proved relatively stable during the 30-day shading treatment (Figure 1E–G). Second, the accumulations of CP43, D1, D2, PsbP, and PsbR were suppressed under HL conditions (0th day) and then progressively increased with shading LL treatment (Figure 1A–D,J,K). Third, the band of target subunit and a band with obviously higher molecular weight (MW) than the target subunit were synchronously detected by CP43 and PsbR antibodies under HL conditions on the 0th day of shading, in which those in CP43 were stoichiometrically adjusted, showing an increase in the higher-molecular-weight (MW) band (~64 kDa) and decrease in the normal band (~43 kDa) (Figure 1A); however, the heavier bands were exclusively detected on the 0th day in PsbR (Figure 1K). Fourth, a band with lower MW than the target subunit was detected by psbN and psbO antibodies at the early stage of shading, in which the PsbN was at a very low level on the 30th day of shading (Figure 1H,I). These suggest that partial subunits of PSII protein complexes in HJY are more vulnerable to HL stress than those in JX, and the HL-induced suppression was reversible under LL conditions.

### 2.2. Effects of HL on Subunits of Light-Harvesting Complexes

HL and shading LL treatments had no significant impact on the accumulation level of PSI light-harvesting complexes I (LHCI) proteins in normal green-leaf cultivar JX (Figure 2). In the LS albino tea cultivar HJY, however, differences in responses of LHCI proteins to HL and shading treatment were observed. The accumulations of Lhca1, Lhca2, Lhca3, and Lhca4 were slightly suppressed under HL conditions (0th day of shading), and the suppression was alleviated with LL shading time in HJY. Until the 30th day of shading, the suppression was completely eliminated, and their expressions were as strong as those in JX (Figure 2A–D). These suggest that LHCI proteins in HJY were more vulnerable to HL stress than those in JX.

The responses of PSII light-harvesting complexes II (LHCII) proteins in HJY to HL obviously differed from those in JX, which remained relatively stable during the whole testing period. HL markedly suppressed the accumulations of Lhcb1, Lhcb2, Lhcb4, Lhcb5, and Lhcb6 in HJY (0th day of shading), in which Lhcb6 was the least affected compared to other LHCII subunits. The suppressions were alleviated progressively under LL conditions by shading, and the bands were as strong as JX on the 30th day of shading (Figure 2E–I). Interestingly, two bands were detected by the Lhcb1 antibody, in which the normal band (~23 kDa) was strongly inhibited by HL and restored under LL conditions by shading, whereas accumulation of the lower MW band (~20 kDa) was inversely induced by HL and disappeared until the 15th day of shading LL treatment (Figure 2E).

The above results show that the accumulations of both LHCI and LHCII complexes in the normal tea cultivar JX were not very affected by HL; however, the LHCI and LHCII subunits in the LS albino tea cultivar HJY were sensitive to HL, in which LHCII was more vulnerable to HL than LHCI.

### 2.3. Effects of HL on Partial Subunits of Electron Transport Complexes

Electron transport complexes subunits of PsaA, PsaD, PetA, PetB, AtpB, AtpC, and AtpH were tested in the present study. HL and artificial shading LL had few effects on accumulations of the tested subunits in JX, with the exceptions of PetB and AtpC. The PetB and AtpC were highly accumulated under HL conditions and then decreased gradually with shading time (Figure 3C,F). In the LS albino cultivar HJY, however, the responses of the tested subunits to light-intensity change were obviously differentiated. First, PetA, AtpB, and AtpH, which were not obviously affected by HL, remained abundantly accumulated during the whole period of testing (Figure 3D,E,G). Second, the levels of PsaD and AtpC were partially suppressed by HL and then increased by progressive shading treatment (Figure 3B,F). Third, PsaA was weakly detected under HL conditions and increased during shading treatment, in which a fragment with lower MW (~37 kDa) was simultaneously observed under HL conditions and disappeared during shading LL period (Figure 3A). Fourth, PetB showed inverse changes compared to the other subunits. It was highly accumulated in the HL but weakened progressively with LL shading time. Furthermore, several bands with MW > 40 kDa were observed besides the normal band (~20 kDa) (Figure 3C).

### 2.4. Effects of Light on PSII Activity

Fv/Fm, the ratio of variable fluorescence to maximum fluorescence, is an indicator reflecting the activity of PSII. Figure 4 shows that the Fv/Fm of JX remained at levels 0.810 to 0.830 during the testing period, but the Fv/Fm of HJY was at a very low level (0.321 ± 0.035) under HL conditions and then increased to a level (0.787 ± 0.030) close to JX until the 15th day of shading (Figure 4), suggesting that PSII activity of HJY was inhibited by HL and could be recovered under LL conditions. The changes in Fv/Fm were consistent with changes in partial PSII proteins and LHC proteins described above, indicating that partial PSII proteins and LHC proteins were involved in the photoinhibition of PSII in HJY.

To further investigate the effects of HL on the photoinhibition of PSII in cultivars HJY and JX, diurnal changes in Fv/Fm on the days of starting and ending artificial shading treatment were determined (Appendix A). It showed that the photosynthetically active radiation (PAR) was at high level during 12 to 14 o’clock on the tested days (Appendix A). However, the albinic leaves of HJY before shading treatment (Appendix A) had markedly lower levels of Fv/Fm than green leaves after shading treatment (Appendix A). The Fv/Fm in both albinic leaves and green leaves of HJY decreased to low levels at 12 or 14 o’clock, and finally, at 18 o’clock in the evening, it was restored to a level as high as that at 8 o’clock. Although the Fv/Fm of green leaves of HJY was as high as the normal cultivar JX at 8 o’clock and 18 o’clock, it was obviously inhibited under HL conditions (1800 μmol/m^2^·s) at noon (12 o’clock) (Appendix A), suggesting that the PSII of HJY was more vulnerable to HL stress than that of JX.

### 2.5. Effects of Inhibitor of Chloroplast-Encoded Protein Synthesis on PSII Activity

Lincomycin is an antibiotic that inhibits the synthesis of chloroplast-encoded proteins [18]. The effects of lincomycin on the PSII activity in leaves of HJY and JX were investigated in the present study. To mimic photoinhibition in vitro, the detached leaves of HJY and JX were kept in the dark with dim light (20 μmol/m^2^·s) overnight and then exposed to a constant light intensity of 1500 μmol/m^2^·s (HL condition). The Fv/Fm was measured in both the presence and absence of lincomycin. In the presence of lincomycin, the repair of PSII was totally blocked, and only the rate of photodamage could be detected, while in the absence of lincomycin, the net rate of photodamage could be balanced by the repair rate. Figure 5 shows that in the absence of lincomycin, after a 1-h exposure to HL, Fv/Fm declined to 0.37 from 0.77 (51.95% decline) in overnight dark-adapted albino leaves of cultivar HJY, while in cultivar JX under the same conditions, the Fv/Fm declined to 0.74 from 0.80 (8.11% decline). Thereafter, the Fv/Fm remained at almost the same levels until after 6 hours of HL exposure (Figure 5). In the presence of lincomycin, however, as the leaves were exposed to HL for 1 hour, the Fv/Fm declined to 0.22 from the dark-adapted value 0.77 in HJY (62.34% decline) and declined to 0.68 from 0.80 (15.00% decline) in JX, and thereafter, the Fv/Fm was further decreased in both JX and HJY. At the end of experiment (6-hour HL exposure), the Fv/Fm was declined to 0.12 (75.32% decline in total) in HJY and to 0.48 (40.00% decline in total) in JX, respectively (Figure 5). These suggest that the de novo biosynthesis and/or repair impairment and photodamage of chloroplast-encoded PSII proteins were involved in the PSII inactivation in HJY, among which photodamage played an important role.

Furthermore, to investigate the relationship of chloroplast-encoded proteins biosynthesis to the foliar albino phenotype in HJY, the albino tea shoots grown under HL conditions (daily sunlight maximum 1800 μmol/m^2^·s) were cut and incubated in 1 mM lincomycin water solution by hydroponics under LL conditions (400 μmol/m^2^·s, 12 h light and 12 h dark) for 15 days, with purified water as control. It showed that the albino tea shoots incubated in water regreened, but those incubated in lincomycin solution remained in the albino phenotype (Appendix A). These suggest that the photodamage of chloroplast-encoded PSII proteins played an important role in the albino phenotype of HJY.

### 2.6. Sequence of Gene PsbC-Encoding CP43 Protein

Figure 1A shows that additional band with higher MW (~64 kDa) than target subunit was detected by CP43 antibody under HL conditions, and the band disappeared with shading treatment for 15 days, accompanying the enhancement of the target band with MW ~43 kDa. Further comparison of CP43 of HJY with several other plant species by immunoblotting showed that the band at ~64 kDa markedly weakened with shading treatment time in HJY, accompanying the enhancement of the band at ~43 kDa, whereas the CP43 subunits from *Arabidopsis thaliana*, *Spinacia oleracea*, and *Solanum lycopersicum* grown under greenhouse conditions had the band exclusively at ~43 kDa (Appendix A). These suggest that the band at ~64 kDa detected by CP43 antibody was a specific response of CP3 to HL stress or a product of photodamaged CP43 in HJY.

Alignments of cDNA nucleotide sequence of PsbC-encoding CP43 and amino acids sequence of CP43 protein in HJY showed that they were totally the same as those in normal tea cultivars JX and LJ-43 (*Camellia sinensis* cv. Longjing 43, a popular cultivar for processing Longjing tea). In addition, the cDNA and amino acid sequences of tea plant were highly identified with *S. lycopersicum* and *S. oleracea* (Appendix A). These indicates that the HL-induced impairment of CP43 in HJY was not induced by the mutation of the PsbC-gene-encoding CP43 protein.

## 3. Discussion

### 3.1. Responses of PSII–LHCII SC Subunits in LS Albino Tea Cultivar to HL Stress

PSII functions as a dimer, which forms PSII–LHCII SC with many pigment-binding LHCB proteins [19]. Each monomer of functional PSII is composed of more than 20 protein subunits and more than 100 cofactors as well as over 1000 bound water molecules [14,20]. The PSII–LHCII SC is a unique membrane protein complex catalyzing light-driven oxidation of water. It converts sunlight into chemical energy, which powers life on the earth [21]. The efficiency of this process is maximized under various environmental conditions by a frequent repair and reassembly cycle following photodamage, many facets of which remain unknown [22]. The transmembrane proteins D1 and D2 form a heterodimer at the PSII RC. The D1 and D2 associate with subunits PsbE, PsbF, and PsbI, which comprise the core RC complex, the smallest PSII–LHCII SC capable of light-induced charge separation. Several membrane proteins such as PsbO, PsbU, PsbV, and PsbQ were included in the functional PSII [14,23]. Surrounding the PSII RC subunits are CP47 and CP43 that bind the Chl a molecule and serve as antennas, harvesting and funneling light energy toward the PSII RC to drive PSII photochemistry [23,24]. PSII is frequently damaged by reactive oxygen species (ROS) arising from the electron-transfer chemistry it performs. D1 and then D2, are damaged and replaced most frequently of all proteins in the complex, while CP43 and CP47 are usually more long-lived [20]. This damage leads to partial disassembly of PSII, replacement of each damaged subunit with a nascent copy, and reassembly of PSII in an intricate process known as the PSII repair cycle [25]. The repair cycle takes place concurrently with de novo synthesis of PSII subunits, which occurs via stepwise PSII assembly from component subcomplexes, in which the PSII RC complex is joined with the CP47 and CP43 precomplexes as well as the membrane-extrinsic proteins [19,25]. Present study shows that the PSI, PSII, LHCI, and LHCII protein subunits as well as PSII activity in the normal green-leaf tea cultivar JX were not vulnerable to HL stress. However, HL induced obvious suppressions on the PSII subunits CP43, D1, D2, PsbP, and PsbR as well as LHCII subunits Lhcb1, Lhcb2, Lhcb4, Lhcb5, and Lhcb6 in HJY, accompanied by a very low activity of Fv/Fm and the albino phenotype. During LL treatment by artificial shading, however, the suppressions of these PSII and LHCII subunits were alleviated with shading time and recovered to levels as high as those of green-leaf cultivar JX, with a concurrent increase in Fv/Fm. HL weakly suppressed the accumulation of PSI complex subunits such as PsaD and LHCI protein subunits Lhca1, Lhca2, Lhca3, and Lhca4. It is concluded that photodamage of PSII plays an important role in the albino phenotype of the LS albino tea plant.

### 3.2. Photodamade of PSII and Albino Phenotype in LS Albino Tea Plant

Besides the photoinhibition or photodamage of PSII, any factors involved in gene mutation, translation, and post-translation modification of genes encoding PSII–LHCII SC proteins would regulate the PSII activity. A single-amino-acid substitution of glutamic acid at position 354 of the CP43 by replacement with glutamine in CP43-Glu354Gln of cyanobacterium *Synechocystis* showed a lower oxygen-evolving activity than its wild-type, accompanied by an altered pattern of flash-dependent delayed luminescence [24]. Translational regulation is a major determinant of gene expression in plastids [26]. The intrinsic mRNA features determine the efficiency of start codon recognition in plastids of higher plants and the efficiency of translation initiation [27]. Changes in mRNA secondary structure might also regulate the translation of PsbA and other plastid genes [28]. HL induced the translation of plastid-encoded PsbA mRNA on the level of translation initiation, resulting in an increase in D1 turnover due to photodamage [29]. The present study showed that the cDNA nucleotide sequences of PsbC and amino acids sequences of CP43 protein encoded by PsbC in HJY were totally the same as those in evergreen tea cultivars JX and LJ43 (Longjing-43, a popular tea cultivar for Longjing tea production), and the sequences of PsbC nucleotides and CP43 amino acids of the tea plant had highly similar identity with *S. oleracea* and *S. lycopersicum*. The sequences of PsbA cDNA and D1 protein as well as PsbD and D2 protein in HJY were also the same as those in cultivars JX and LJ43 (data not shown). The same phenomenon was observed in other vascular plants: neither transcription nor translation of PsbD-encoding D2 and PsbC-encoding CP43 are rate-limiting for PSII biogenesis [30]. It is concluded that it is not the mutation of gene PsbC that leads to the block of CP43 protein accumulation and inactivation of PSII in LS albino tea cultivar HJY.

Post-translation modifications of PSII–LHCII SC subunits are important factors regulating the activity of PSII. D1 maturation was found to play a key role during PSII–LHCII SC assembly, and the loss of D1 maturation affects the incorporation of CP43 during PSII–LHCII SC formation [31,32]. After maturation of the D1 protein by the luminal C-terminal processing protease CTPA [33], an “RC47 subcomplex” is formed by binding of CP47 [34] and rapid addition of PsbH, PsbR, and PsbTc [35]. Finally, CP43, PsbK, and PsbZ bind to the PSII RC [35,36]. Immunoblotting in the present study shows that the impairments of D1, D2, and CP43 in HJY were exclusively observed in the HL-induced albinic leaves, and under LL conditions, they were as normal as those in evergreen tea cultivars JX and other plant species, such as *A. thaliana*, *S. lycopersicum*, and *S. oleracea*. Two bands were detected at MW ~64 kDa and ~43 kDa by the CP43 antibody in the albinic leaf of HJY, respectively. There might be two possibilities for this phenomenon. First, the ~43 kDa band was the functional CP43, and the ~64 kDa band was the crosslinking product of degraded CP43 fragments induced by photodamage. Second, the ~43 kDa band was the functional mature CP43, and the ~64 kDa band was the primitive form of CP43. Previous studies showed that the green leaves grown under LL conditions were bleached when exposed to HL conditions and vice versa [4,5]. If the latter assumption was true, it would mean that the maturation of primitive CP43 took place under LL conditions, and on the contrary, the mature CP43 in leaves grown under LL conditions would be reversed to primitive CP43 when exposed to HL. However, this is very unlikely because no evidence has been shown confirming the reversion from mature CP43 to premature CP43 so far.

Photoinhibition is an imbalance between PSII photodamage and its subsequent repair [37], in which rapid degradation and replacement of PSII proteins takes place [29]. D1 is a very rapidly turning-over protein and has a faster translation rate than other subunits in PSII [38]. PSII photodamage is involved in disintegration of the Mn center in PSII, which then leads to an energy imbalance and an ill-defined oxidative damage to D1 protein fragments, leading to inactivation of PSII and cleavage of the D1 subunit by proteases, and PSII activity inhibition is the insufficient de novo biosynthesis of D1 to repair PSII under environmental stresses [39]. Lincomycin is an antibiotic suppressing the biosynthesis of chloroplast-encoded proteins [17]. Fv/Fm, an indicator of PSII activity, could be used to assess the photoinhibition induced by de novo biosynthesis and/or repair of PSII in the presence and absence of lincomycin. The presence of lincomycin totally blocked the de novo synthesis and repair of PSII subunits, leading to a 84.44% decline of Fv/Fm in HJY, whereas a 51.95% decline of Fv/Fm was observed in the absence of lincomycin, where the net rate of photodamage could be balanced by the de novo synthesis and repair rate, suggesting that the 32.49% inactivation of PSII activity was induced by impairment of de novo synthesis and/or repair of PSII subunits in HJY. The albino leaves of HJY regreened under LL conditions, but lincomycin blocked the regreening, suggesting that de novo biosynthesis of PSII is involved in the albino phenotype in HJY. In JX, however, the presence of lincomycin led to a 40.00% decline of PSII activity, and in the absence of lincomycin, only an 8.80% decline of PSII activity was observed, suggesting that the 31.20% decline of PSII activity was caused by impairment of the de novo biosynthesis and/or repair of PSII in JX. These suggest the inhibitive effects of HL on the de novo biosynthesis of PSII subunits were at the same level in both HJY and JX, but suppression of HL on the repair of PSII was more serious in HJY than in JX. These findings lead to a conclusion that the impairment in repair of the photodamaged PSII plays a key role in the inactivation of PSII activity and albino phenotype in HJY.

### 3.3. Multicopy of Photosynthetic Proteins and Adaptation to HL Stress in LS Albino Tea Plant

A typical PSII–LHCII SC in plants is formed by four trimers of Lhcb proteins (LHCII trimers), which are bound to the PSII RC core dimer via monomeric antenna proteins [40]. The Lhcb1 protein is usually encoded by several nuclear *Lhcb1* genes [41], and its copy number may be up to 16 [42]. Individual *Lhcb1* genes encode apoproteins differing with regard to the apparent MW ranging from 24.8 kDa to 25.0 kDa [43]. An individual Lhcb1 molecule is assumed to bind eight Chl a, six Chl b, four carotenoid, and two lipid molecules [44]. The ligands of 14 Chls include seven amino-acid residues, two backbone carbonyls, four water, and one phosphatidylglycerol molecule [44]. Wheat cultivar “BN207” accumulated a higher level of Lhcb1 protein than its parents, resulting in improvement of photosynthetic efficiency by promoting light energy absorption and conversion [45].

The multicopy of partial subunits of PSII–LHCII SC is an adaptation to the environmental stress conditions. In spinach, there are three copies of Lhcb1, and about the same relative amounts of the three transcripts are present under normal light conditions. However, in the leaves exposed to 40 h of white light, a differential transcript accumulation occurred in which the accumulation of Lhcb1.3 was 10-fold of Lhcb1.1 and Lhcb1.2 [46]. In *Prochlorococcus*, the HL-adapted ecotype MED4 has more genes encoding putative HL-inducible proteins (HLIP) and photolyases to repair UV-induced DNA damage, whereas LL-adapted ecotype MIT 9313 possesses more genes associated with the photosynthetic apparatus [47]. The LL strains of *Prochlorococcus* have multiple-copy *pcb* genes encoding the major chlorophyll-binding and light-harvesting antenna proteins, but the HL strains have a single-copy *pcb* gene [48]. In rice, there are two copies of the *Lhcb1* gene (*Lhcb1a* and *Lhcb1b*). A rice mutant “dye1-1” (delayed yellowing1-1) with higher level of Lhcb1a and Lhcb1b than the wild-type accumulated high levels of Chls in the pre-senescent leaves and showed a stay-green phenotype. The increase in the Lhcb proteins was found to be regulated at the protein level instead of the mRNA level [49]. Dephosphorylation of D1 protein is a prerequisite for D1 degradation in vivo [50]. PsbO plays a key role in regulating D1 dephosphorylation and/or degradation in plants. An in vitro experiment in spinach showed that PsbO functioned as a “chaperone” in preventing D1 aggregation [51]. Guanosine triphosphate (GTP) stimulates the dissociation of PsbO from PSII under HL conditions known to also release Mn^2+^ and Ca^2+^ ions from the oxygen-evolving complex and to induce degradation of the PSII RC D1 protein [52]. Sequencing of the *Arabidopsis* genome (AGI, 2000) showed that there were two PsbO genes, i.e., At5g66570 encoding PsbO1 and At3g50820 encoding PsbO2. The function of PsbO1 is mostly to enhance PSII activity, whereas the interaction of PsbO2 with PSII regulates the HL-induced turnover of the D1 protein, increasing its accessibility to the phosphatases and proteases involved in its degradation [53]. However, only one PsbO protein was characterized in spinach [54]. In the present study, two copies of PsbN, PsbO, and Lhcb1 were detected in HJY under HL conditions, in which the copies with lower MW (PsbN2, PsbOb, and Lhcb1b) were almost completely degraded in shading time up to 15 days, accompanied by accumulation of the copies with higher MW ones (PsbN1, PsbOa, and Lhcb1a). The ~17 kDa band found in the HL-stressed PsaA was markedly smaller than the normal PsaA (~52 kDa), suggesting it was a degraded fragment of PsaA instead of its isomer. Multicopy of the isomers were not detected in JX and in leaves of HJY after the 30-d shading treatment. It is concluded that the multicopy of photosynthetic protein subunits plays a specific role in the LS albino tea cultivars in adaptation to different light conditions.

### 3.4. HL-Induced Degradation of PSII Subunits and Crosslinking of Their Degradation Fragments

The PSII RC proteins D1 and D2 are normally shielded by CP43 and CP47 subunits, respectively, and they are the main targets for photodamage [55]. The damaged PSII RC proteins are selectively degraded by protease, in which the protease accessibility induced by PSII disassembly is an important determinant in the selection of the D1 and D2 subunits to be degraded by the proteases. In a photochemically active PSII complex lacking CP43, the D1 becomes exposed and then is selectively degraded by protease even in the dark. In addition, removal of the CP47 subunit will increase accessibility of proteases to the D2 subunit, resulting in degradation of D2 [55]. The Deg protease accesses photodamaged D1, resulting in D1 fragmentation, and subsequently, the degraded D1 fragments undergo processive degradation by protease FtsH, which is called a two-step model [56]. The CP43 contains three β-carotene (β-Car) and five Chl *a* molecules, which are involved in the energy transfer from β-carotene to Chl *a*. These five Chl *a* molecules bleached more readily than others under constant illumination. The CP43 is more easily damaged by illumination than CP47, and the secondary structures, especially the helix of CP43, were altered upon illumination [57]. Concomitant with the damage to the D1, D2, and CP43, some aggregates with higher MW were formed by crosslinking between the degradation fragments and/or D1 and CP43 subunits [51]. Degradation of the damaged D1 and its replacement by nascent D1 are at the heart of a PSII repair cycle. In mature plant chloroplasts, light stimulates the recruitment of ribosomes specifically to PsbA mRNA to provide nascent D1 for PSII repair [58]. The band with higher MW than target subunits, such as CP43, D2, PsbR, and Lhcb5, should be the crosslinked products of corresponding PSII subunits and/or their degradation fragments. Previous study showed that three fragments with MW 17.0, 15.5, and 14 kDa from degradation of CP43, two fragments with MW 17.0 and 24 kDa from degradation of D1, and one fragment with MW 29 kDa from degradation of D2 were identified in HL-stress spinach, respectively [51]. Concomitant with the damage to the D1, D2 and CP43, crosslinked products of the degraded CP43 proteins were formed, which were identified as slow-moving, smeared bands in the higher MW range on the electrophoresis gel [51]. Present study shows that the MW of crosslinking products detected by anti-CP43, anti-PsbR, and anti-Lhcb5 ranged from ~52 kDa to ~60 kDa, suggesting they were trimer or tetramer crosslinking products. It is concluded that HL-induced degradation of thylakoid membrane proteins involves in the photodamage of PSII and foliar albino phenotype in HJY, in which the HL-induced degradation protein fragments were partially aggregated by crosslinking, resulting in the formation of higher MW inactivation proteins.

Based on the present results and the available knowledge relating to the HL-induced PSII assembly, photodamage, and repair, we conclude that HL-induced disassembly of PSII involves the PSII inactivation and foliar albino in LS albino tea plants. Under LL conditions, D1 and D2 are not accessed by proteases Deg and HtsH because they are shielded by subunits CP43, CP47, PsbO, PsbP, PsbR, Lhcb1, Lhcb2, Lhcb4, Lhcb5, and Lhcb6. Under HL conditions, however, the D1 and D2 are exposed to Deg and FtsH owing to the partial degradation or detachment of subunits CP43, CP47, PsbO, PsbP, PsbR, Lhcb1, Lhcb2, Lhcb4, Lhcb5, and Lhcb6, resulting in the inactivation of D1 and D2 as well as disassembly of PSII–LHCII SC, which results in the decrease in chlorophylls concentration or the albino phenotype. This process is reversible under LL conditions where the PSII–LHCII SC is reassembled using the repaired or de novo synthesis subunits of PSII (Figure 6).

Under LL conditions, D1 and D2 are not accessed by proteases Deg and FtsH because they are shielded by the other PSII–LHCII SC subunits in the matured PSII–LHCII SC (the lower left). HL induces partial degradation or detachment of subunits CP43, CP47, PsbP, PsbR, Lhcb2, Lhcb4, Lhcb5, and Lhcb6, causing D1 and D2 to be exposed to Deg and FtsH, resulting in the degradation and inactivation of D1 and D2 as well as disassembly of PSII–LHCII SC (the upper left). As the illumination is reversed to LL (on the right), partially photodamaged PSII–LHCII SC subunits are repaired, and then, the damaged PSII–LHCII SC is reassembled using the repaired or de novo synthesis subunits.

Green, matured and functional subunits; light green, repaired or de novo synthesized subunits; light red or pink, lightly photodamaged subunits; deep red, seriously photodamaged subunits. 

## 4. Materials and Methods

### 4.1. Plant Materials

A LS albino tea cultivar HJY (*Camellia sinensis* cv. Huangjinya) and normal evergreen tea cultivars JX (*Camellia sinensis* cv. Jinxuan) and LJ-43 (*Camellia sinensis* cv. Longjing-43) were used as tested materials in the present study. They were all ten-year-old plants grown under the same agricultural practices in a commercial field of the Daqing Tea Farm (Hangzhou, China, latitude: 30.05° N, longitude: 119.87° E).

Young plants of *Arabidopsis thaliana*, *Spinacia oleracea*, and *Solanum lycopersicum* were grown under high-light conditions before sampling.

To investigate the effects of sunshine shading on the albino phenotype, black polyethylene sunshade nets with approximately 30% transmittance were used to shield the plants at 50 cm above the plucking table of the tea bushes in the early July 2018. The fourth leaf beneath the apex bud was sampled for determining chloroplast proteins and cloning the cDNA of PsbC. The leaves were sampled on the 0th (the day of shading treatment), 6th, 15th, and 30th days after shading treatment. The sampled leaves were wrapped in aluminum foil, frozen in liquid nitrogen, and then stored at −80 °C until use.

### 4.2. Determination of Fv/Fm

Fv/Fm was determined using a fluorimeter (Handy PEA, Hansatech Instruments Ltd., Norfolk, UK) according to the instructions after 30 min dark adaptation. The fourth leaves beneath the apex bud on the tea bushes were chosen for the in vivo Fv/Fm test, in which 10 leaves from 5 plants for each treatment were determined.

For lincomycin treatment experiments, the fourth leaves beneath the apex bud were detached from shoots of tea plants after 30-day shade acclimation. The detached leaves were harvested at 19:00, kept in 20 μmol/m^2^·s dim light with the petioles in 1 mM lincomycin solution for 12 hours to exclude the effect of de novo chloroplast-encoded protein synthesis on the susceptibility of leaves to photoinhibition, and then floated with adaxial side up in the same solution and illuminated at a photon flux density of 1500 μmol/m^2^·s at 25 °C (HL conditions). For the control, the lincomycin solution was replaced by water, with the same dim light adaption and illumination as the one in the lincomycin solution. Fv/Fm was determined as previous method [18] after 0, 1, 2, 3, 4, 5, and 6 h of HL illumination, with 10 leaves in each treatment.

### 4.3. Protein Preparation

Intact chloroplasts were isolated in isolation buffer (0.33 M sorbitol and 20 mM HEPES/KOH, pH 7.6) and then osmotically ruptured in 20 mM HEPES/KOH (pH 7.6). Thylakoid proteins were separated from the thylakoid membranes using centrifugation, as described by Cai et al. [18]. The chlorophyll levels were measured according to Cai et al. [18].

### 4.4. SDS-PAGE and Immunoblotting Analysis

For Western blot analysis, the proteins in the thylakoid membranes were solubilized and separated by commercial linear-gradient SDS polyacrylamide gels (Shaanxi ZHHC Bio-medical Technology Co., Ltd., Xi’an City, China), including 12% universal PAGE for most subunits and 16% special PAGE for detecting PsbE, PsbH, PsbI, PsbN, PsbR, PsaD, and AtpH because of their lower molecular weights. The proteins were transferred onto nitrocellulose membranes (0.2 μm, GE) using the tank transfer systems offered by Mini Trans-Blot (Bio-Rad Laboratories (Shanghai) Co., Ltd., Shanghai, China) and probed with 27 specific primary antibodies (commercial products from Agrisera (Vännäs, Sweden) and kindly supplied by Dr Lixin Zhang). All the antibodies were diluted in 1:2000. Signals from the secondary conjugated antibodies were detected by the enhanced chemiluminescence method using Fujifilm LAS3000 according to the published methods [18]. The band intensities were quantified with Image J [59].

### 4.5. Cloning and Sequencing of PsbC

The tested leaf (100 mg) was ground with liquid nitrogen, and total RNA was extracted with Biofit Plant RNA Extraction Kit ver. 1.5 (Chengdu Biofit Biotechnologies Co., Ltd., Chengdu, China). According to the instruction of Goldenstar RT6 cDNA Synthesis Kit ver. 2 (Tsingke Biotechnology Co., Ltd., Beijing, China), the extracted RNA was incubated with DNase to remove the genomic DNAs and then used to synthesize the first cDNA strand. The synthesized cDNA and the primers (forward: 5′-TTGGATGGCGGCTCAAGATC-3′, reverse: 5′-GGATTCCTACTTCAAACATTGGATCTC-3′) were used to clone the PsbC by using Mix(green) PCR kit (Tsingke Biotechnology Co., Ltd., Beijing, China). The PCR were performed under the following thermal cycle conditions: 1 cycle pre-denaturation at 98 °C for 2 min, 35 cycles of denaturation at 98 °C for 10 s, annealing at 58 °C for 10 s and extension at 72 °C for 30 s, followed by a final extension step at 72 °C for 5 min. The amplified PsbC PCR product was bi-directionally sequenced by Sanger methods [60], and the full length of the PsbC was obtained through the overlapping of the reads. The ORF of the PsbC was predicted and deduced to amino acids by the Editseq software ver. 7.1.0 in DNAStar package (DNAStar Inc.). Reference sequences of PsbC from *C. sinensis* cv. Longjing 43 (GenBank accession No. KF562708.1, position 35705-37126, accessed on 6 November 2013), *Solanum lycopersicum* (GenBank accession No. DQ347959.1, position 34591-36012, accessed on 13 January 2014), and *Spinacia oleracea* (GenBank accession No. NC_002202.1, position 32516-33937, accessed on 15 Aprirl 2009) were downloaded from GenBank (www.ncbi.nlm.nih.gov/, accessed on 28 June 2022). The multiple sequence alignments of nucleotides and amino acids were performed on the GenDoc (www.psc.edu/biomed/genedoc, accessed on 28 June 2022).

## Figures and Tables

**Figure 1 ijms-23-08522-f001:**
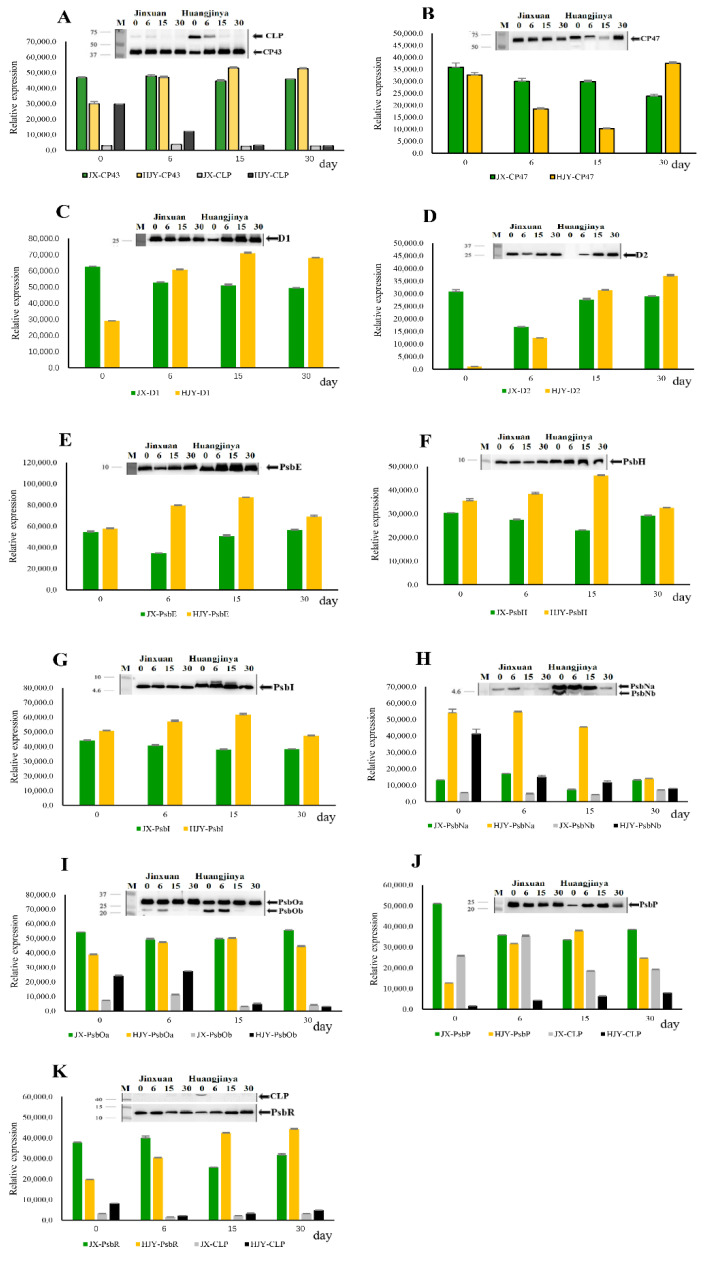
(**A**–**K**) Immunoblotting of PSII subunits during shading treatment. Thylakoids were extracted from the 4th leaf beneath apex bud on tea shoots, and 1 µg of chlorophyll was used each lane. The blots were probed with specific antibodies: anti-CP43, anti-CP47, anti-D1, anti-D2, anti-PsbE, anti-PsbH, anti-PsbI, anti-PsbN, anti-PsbO, anti-PsbP, and anti-PsbR. CLP, crosslinking product. The data 0, 6, 15, and 30 refer to the days of shading treatment. The bars are expressed as mean ± standard deviation (*n* = 3).

**Figure 2 ijms-23-08522-f002:**
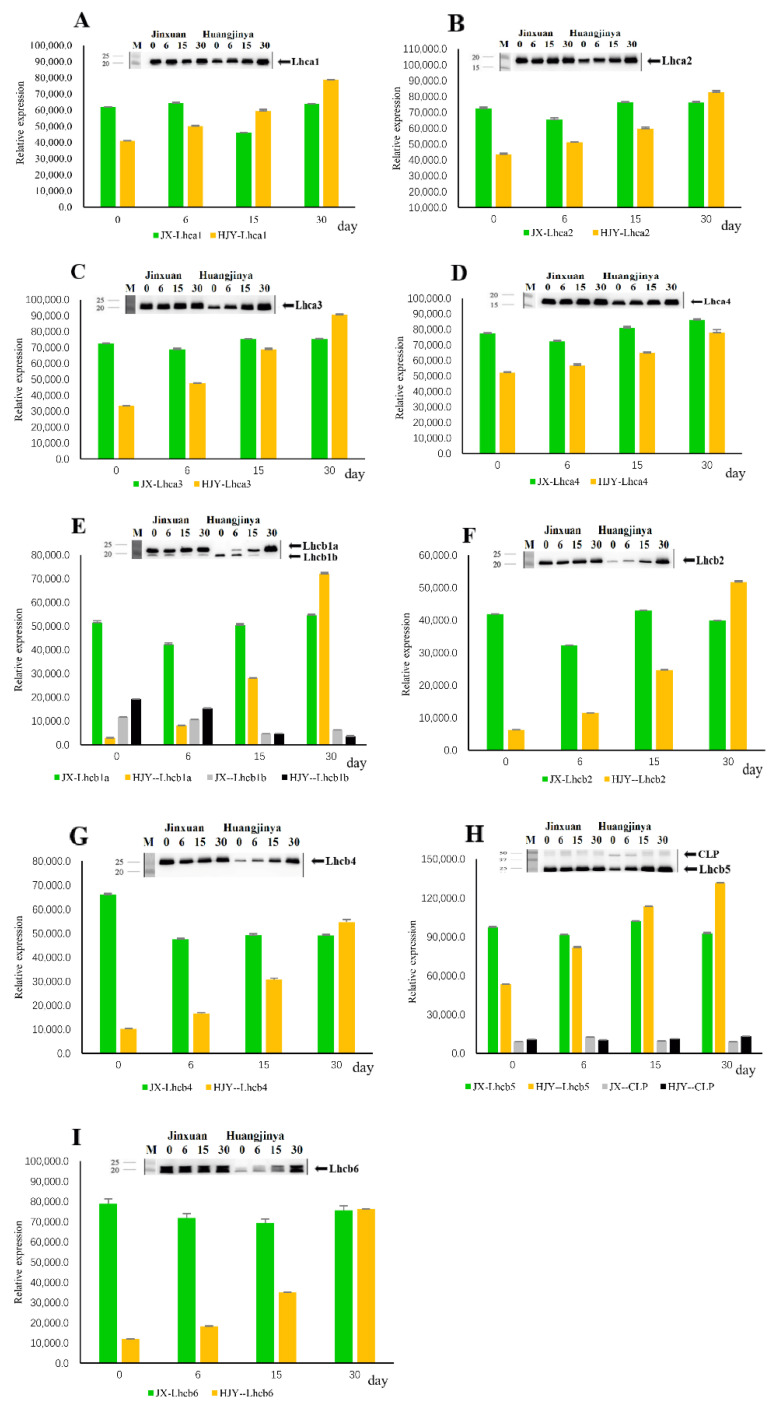
(**A**–**I**) Immunoblotting of LHC proteins during shading treatment. Thylakoids were extracted from the 4th leaf beneath apex bud on tea shoots, and 1 µg of chlorophyll was used each lane. The blots were probed with specific antibodies: anti-Lhca1, anti-Lhca2, anti-Lhca3, anti-Lhca4, anti-Lhcb1, anti-Lhcb2, anti-Lhcb4, anti-Lhcb5, and anti-Lhcb6. CLP, crosslinking product. The data 0, 6, 15, and 30 refer to the days of shading treatment. The bars are expressed as mean ± standard deviation (*n* = 3).

**Figure 3 ijms-23-08522-f003:**
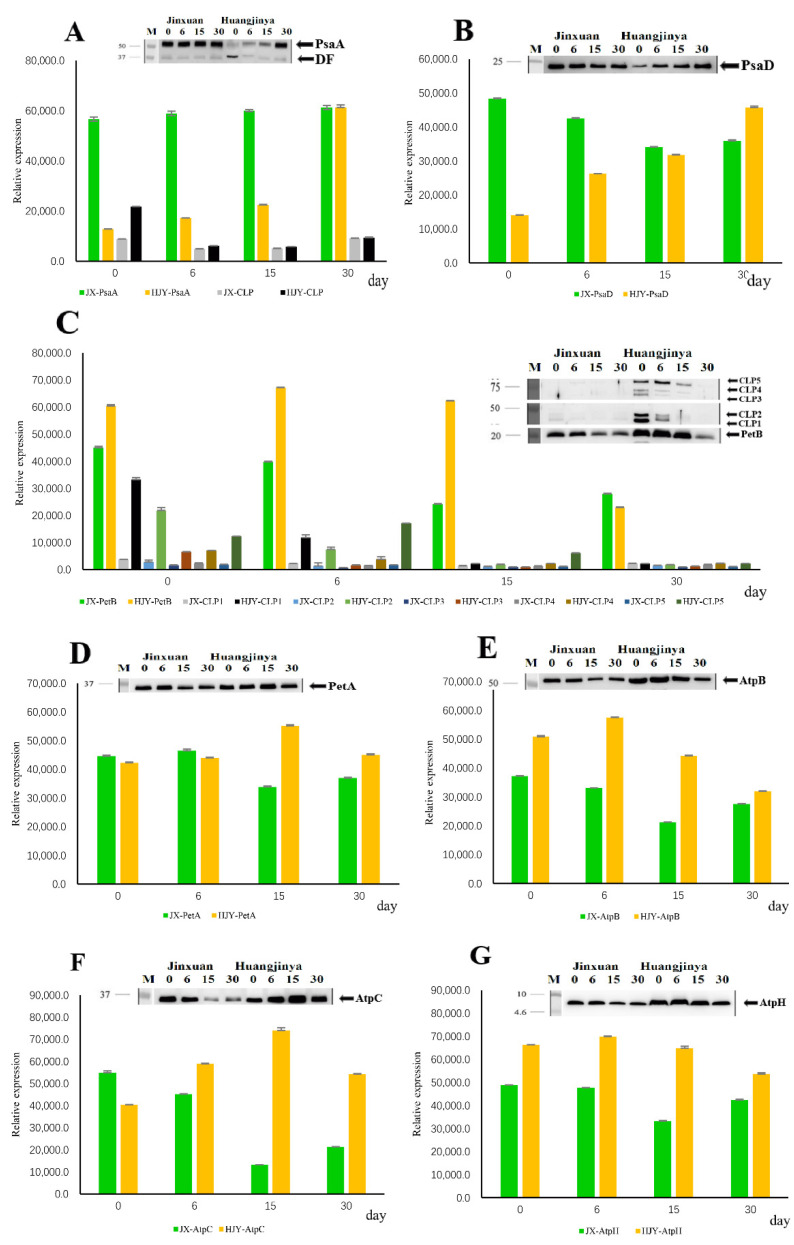
(**A**–**G**) Immunoblotting of partial subunits of electronic transport complexes. Thylakoids were extracted from the 4th leaf beneath apex bud on tea shoots, and 1 µg of chlorophyll was used each lane. The blots were probed with the following specific antibodies: anti-PsaA, anti-PsaD, anti-PetB, anti-PetA, anti-AtpB, anti-AtpC, and anti-AtpH. CLP, crosslinking product; DF, degradation fragment. The data 0, 6, 15, and 30 refer to the days of shading treatment. The bars are expressed as mean ± standard deviation (*n* = 3).

**Figure 4 ijms-23-08522-f004:**
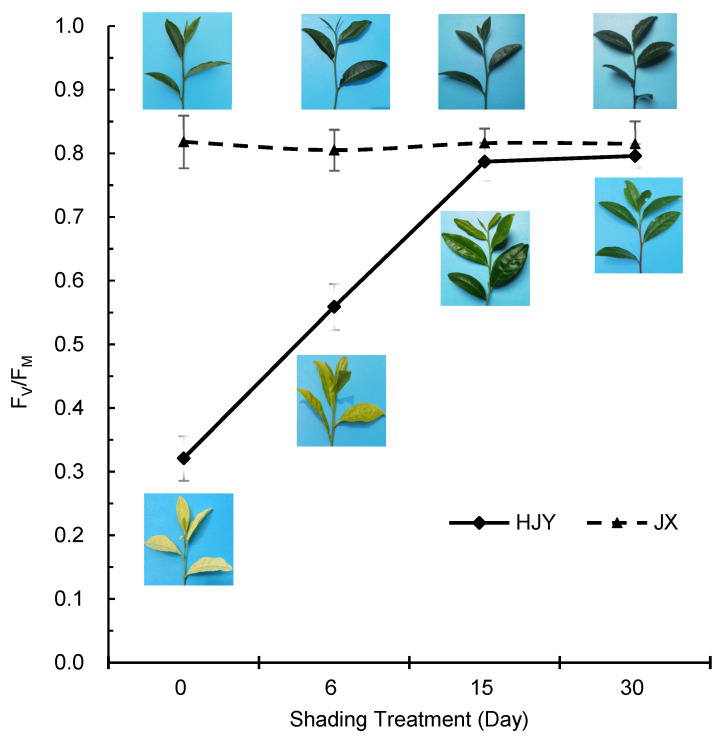
Changes in Fv/Fm during shading treatment period. The Fv/Fm was determined on the 4th leaf beneath apex bud, and 10 leaves were tested each treatment. HJY, cultivar “Huangjinya”; JX, cultivar “Jinxuan”.

**Figure 5 ijms-23-08522-f005:**
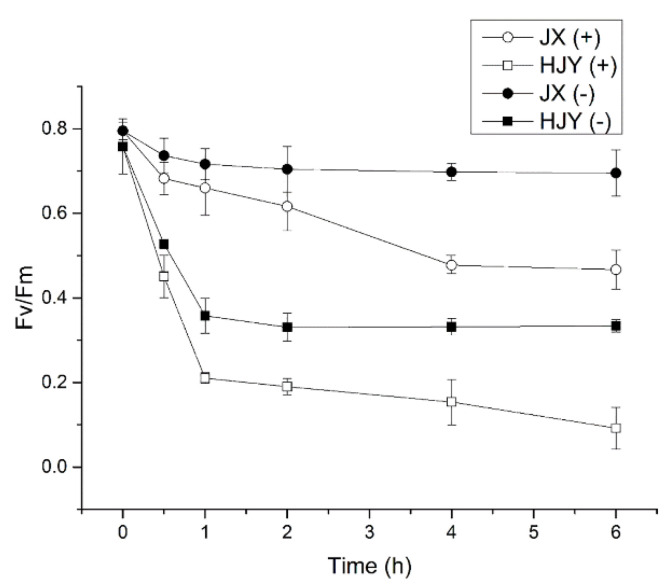
Effects of lincomycin on Fv/Fm in leaves of HJY and JX. Fv/Fm was measured for detached leaves (the 4th leaf beneath apex bud on bush under 30-day shading treatment) in the absence (–) or presence (+) of lincomycin during exposure to an irradiance of 1500 μmol m^−2^ s^−1^. The leaves were collected on 30th August 2018 at dusk and immersed in water before photoinhibition. Time 0 represents the dark-adapted samples before light exposure. For lincomycin treatment, the leaves were kept in darkness with the petioles in 1 mM lincomycin solution for 12 h before light exposure. The values are the mean ± standard deviation from at least six replicated experiments. HJY, cultivar “Huangjinya”; JX, cultivar “Jinxuan”.

**Figure 6 ijms-23-08522-f006:**
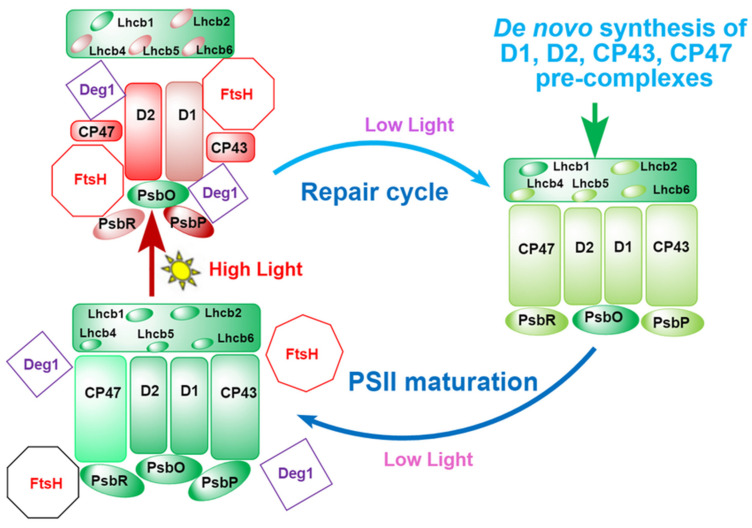
HL-induced disassembly of PSII–LHCII SC and its reassembly under LL conditions in LS albino tea plant.

## Data Availability

Not applicable.

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
