# Peer review of "High-Light-Induced Degradation of Photosystem II Subunits’ Involvement in the Albino Phenotype in Tea Plants"

_ijms, 2022, doi:10.3390/ijms23158522_

Round 1

Reviewer 1 Report

The prepared manuscript is proteomic in nature and, while interesting, has a few shortcomings.

1. I do not understand why Arabidopsis is added as a control on some blots. If we add something, it should be used in all photos on a given panel.

2. Does not understand why plants such as spinach and Arabidopsis are grown in high light as if these were the control conditions.

3. Nice blots, really, but why take up entire panels - I don't want to see one representative, prettiest blot - I want bars from densitometric measurements with applied deviations and calculated statistics.

Pictures can be placed in supplements – or, better, make panels, where there are cut-out strips of protein, and then the picture size will be significantly reduced - and this would be part A, and in part B, show a bar chart from quantitative measurements.

4. In the methodology in the part of protein preparation, something is lost. There is an additional period and "The" - please complete it.

5. In the section on electrophoresis and transfer, there is no information about the percentage of acrylamide gel and its producer, no information about the transfer membrane's producer and the transfer type - at least insert a reference to the entire method. Please state the antibody numbers and the dilutions used.

The work is interesting, but because there are many changes concerning the presentation of the most important results, I rate it a major revision.

Author Response

Responses to Reviewer-1’s comments

    Thank Reviewer 1 for your valuable comments. We revised the manuscript according to the comments and the revised version is submitted for your consideration for publication in IJMS.

    The point to point responses to the comments are as follows:

  1. I do not understand why Arabidopsis is added as a control on some blots. If we add something, it should be used in all photos on a given panel.

    Authors’ response to the comment:

    Thank reviewer-1’s comment.

    We firstly found there were two different forms detected by CP43 antibody in albino leaves of tea plants while only one in Arabidopsis thaliana, and then we detected D1, D2, CP47 of PSII to find their differences. However, our purpose for Figure 1 was designed for immunoblotting of PSII subunits during shading treatment in the leaves of light sensitive tea plants. Therefore, the lanes of Arabidopsis thaliana were not included in Figure 1 in the revised version. And the Figure legend was correspondingly changed as “The data 0, 6, 15 and 30 refers to the days of shading treatment.

  1. Does not understand why plants such as spinach and Arabidopsis are grown in high light as if these were the control conditions.

    Authors’ response to the comment:

    We are sorry for the mis-presentations. The plants of Arabidopsis thaliana, Spinacia oleracea and Solanum lycopersicum tested in the present study were grown in green house without shading treatment. The legend for Figure S4 should be presented as following:

    Figure S4. Immunoblotting of CP43 from various plant species

M, marker; 1, HJY grown under natural high light condition; 2, HJY under natural relatively low light condition; 3, Arabidopsis thaliana grown in green house condition; 4, Spinacia oleracea grown in green house condition; 5, Solanum lycopersicum grown under in green house condition.

  1. Nice blots, really, but why take up entire panels - I don't want to see one representative, prettiest blot - I want bars from densitometric measurements with applied deviations and calculated statistics.

    Pictures can be placed in supplements – or, better, make panels, where there are cut-out strips of protein, and then the picture size will be significantly reduced - and this would be part A, and in part B, show a bar chart from quantitative measurements.

    Authors’ response to the comment:

    Thank you very much for your valuable comments. Quantitative data was obtained by scanning the bands with Image J. and the data in Figures 1-3 were presented as bar charts with corresponding WB bands according to your comments.

  1. In the methodology in the part of protein preparation, something is lost. There is an additional period and "The" - please complete it.

    Authors’ response to the comment:

    We are very sorry for the mistakes. The full sentence should be as: The chlorophyll levels were measured according to Cai et al.18.

  1. In the section on electrophoresis and transfer, there is no information about the percentage of acrylamide gel and its producer, no information about the transfer membrane's producer and the transfer type - at least insert a reference to the entire method. Please state the antibody numbers and the dilutions used.

    Authors’ response to the comment:

    Thank you for your valuable comments. The related information was included in the Section “SDS-PAGE and immunoblotting analysis” of the revised version.

    For western blot analysis, the proteins in the thylakoid membranes were solubilized, separated by commercial linear-gradient SDS polyacrylamide gels (Shaanxi ZHHC Bio-medical Technology Co., Ltd, Xi’an City, China), including 12% universal PAGE for most subunits, and 16% special PAGE for detecting PsbE, PsbH, PsbI, PsbN, PsbR, PsaD, and AtpH because of their lower molecular weights. The proteins were transferred onto nitrocellulose membranes (0.2μm, GE) using the tank transfer systems offered by Bio-Rad (Mini Trans-Blot), and probed with 27 specific primary antibodies (commercial products from Agrisera (Vännäs, Sweden) and kindly supplied by Dr Lixin Zhang). All the antibodies were diluted in 1:2000. Signals from the secondary conjugated antibodies were detected by the enhanced Chemiluminescence method using Fujifilm LAS3000 according to the published methods 18.

The work is interesting, but because there are many changes concerning the presentation of the most important results, I rate it a major revision.

    Authors’ response to the comment:

    We revised the manuscript and the changes were marked in red ink in the revised manuscript.

Reviewer 2 Report

Comments

The author of the manuscript entitled “High light induced degradation of Photosystem II subunits involvement in the albino phenotype in tea plants” can be useful for wide range of readers in the journal. There are some serious issues on where authors need to concentrate.

The position of figure 7 is wrong. Adjust is in the result section.

The description of Figure 7 is missing in the text.

Citations are not in serial order in the text. Check it

Example—Line 272 to 283.

Try to avoid using figure and table numbers in the discussion section. They are already mentioned in the result section.

Example…line 299, 302, 303, 414, 443, 468.  Etc.

Figure 1 , 2 3: A-K……. images need to be replaced them, as the numbering are not clear

Author Response

Responses to Reviewer-2’s comments

Thank Reviewer 2 for your valuable comments. We revised the manuscript according to the comments and the revised version is submitted for your consideration for publication in IJMS.

The point to point responses to the comments are as follows:

Comments

1.The author of the manuscript entitled “High light induced degradation of Photosystem II subunits involvement in the albino phenotype in tea plants” can be useful for wide range of readers in the journal. There are some serious issues on where authors need to concentrate.

     The position of figure 7 is wrong. Adjust is in the result section.

Authors’ response to the comment:

Thank you for your comment. We checked the manuscript and found the “Figure 7” should be replaced by “Figure 6”, because one of the previous Figure was transferred to supplement data.

2.The description of Figure 7 is missing in the text.

Authors’ response to the comment:

As previous response, the “Figure 7” should be replaced by “Figure 6”.

3.Citations are not in serial order in the text. Check it, Example—Line 272 to 283.

Authors’ response to the comment:

Thank you for your comment. We checked the manuscript and the citations were numbered in serial order in the revised manuscript, and some additional references were also cited in the necessary discussions.

4.Try to avoid using figure and table numbers in the discussion section. They are already mentioned in the result section. Example…line 299, 302, 303, 414, 443, 468.  Etc.

Authors’ response to the comment:

The Figure and Table numbers in the discussion section were delleted in the revised version.

5.Figure 1 , 2 3: A-K……. images need to be replaced them, as the numbering are not clear

 Authors’ response to the comment:

Thank you very much for your valuable comments. According to your comments, the data in Figures 1-3 were presented as bar charts with corresponding WB bands, in which the quantitative data were obtained by scanning the bands using Image J.

We revised the manuscript and the changes were marked in red ink in the revised manuscript.

Round 2

Reviewer 2 Report

There are some minor issues in the manuscript. 

Line 473 and 488: no need to write double “methods”

Line 502: which “previous method”?. Mention it.

Line 535:  references missing and wrong citation number “Sanger methods (82),”

Line 538: Write in italic form “C. sinensis”

Author Response

Thank Reviewer 2 for your kind comments. We revised the manuscript according to the comments. The major changes are as follows:

1.Line 473 and 488: no need to write double “methods”.

    Authors'  response to the comment:  The "Methods" in Line 488 was delleted.

2.Line 502: which “previous method”?. Mention it.

Authors'  response to the comment:  The "Previous methods" refers to that described in reference 18. The reference 18 waas cited. 

3.Line 535:  references missing and wrong citation number “Sanger methods (82),”

Authors'  response to the comment:  The citation number was wrong and the correct reference was cited as reference 61 in the revised version.

4.Line 538: Write in italic form “C. sinensis”.

Authors'  response to the comment: The "C.  sinensis" was rewriten in italic form in the revised version.